# Coaching Belgian and Dutch Broiler Farmers Aimed at Antimicrobial Stewardship and Disease Prevention

**DOI:** 10.3390/antibiotics10050590

**Published:** 2021-05-17

**Authors:** Nele Caekebeke, Moniek Ringenier, Franca J. Jonquiere, Tijs J. Tobias, Merel Postma, Angelique van den Hoogen, Manon A. M. Houben, Francisca C. Velkers, Nathalie Sleeckx, Arjan Stegeman, Jeroen Dewulf

**Affiliations:** 1Veterinary Epidemiology Unit, Department of Reproduction, Obstetrics and Herd Health, Faculty of Veterinary Medicine, Ghent University, Salisburylaan 133, 9820 Merelbeke, Belgium; moniek.ringenier@ugent.be (M.R.); merel.postma@ugent.be (M.P.); jeroen.dewulf@ugent.be (J.D.); 2Division Farm Animal Health, Department of Population Health Sciences, Faculty of Veterinary Medicine, Utrecht University, Yalelaan 7, 3584 CL Utrecht, The Netherlands; franca@jonquiere.nl (F.J.J.); t.j.tobias@uu.nl (T.J.T.); a.vandenhoogen@westfort.nl (A.v.d.H.); f.c.velkers@uu.nl (F.C.V.); j.a.stegeman@uu.nl (A.S.); 3GD Animal Health, Arnsbergstraat 7, 7418 EZ Deventer, The Netherlands; m.houben@gddiergezondheid.nl; 4Experimental Poultry Centre, Poiel 77, 2440 Geel, Belgium; nathalie.sleeckx@provincieantwerpen.be

**Keywords:** antimicrobial use, biosecurity, coaching, farmers’ attitude, poultry production

## Abstract

A reduction in antimicrobial use (AMU) is needed to curb the increase in antimicrobial resistance in broiler production. Improvements in biosecurity can contribute to a lower incidence of disease and thereby lower the need for AMU. However, veterinary advice related to AMU reduction or biosecurity is often not complied with, and this has been linked to the attitudes of farmers. Behavior change promoted by coaching may facilitate uptake and compliance regarding veterinary advice. Thirty broiler farms in Belgium and the Netherlands with high AMU were included in this study for 13 months. For each farmer, the attitude towards AMU reduction was quantified using an adjusted Awareness, Desire, Knowledge, Ability, and Reinforcement (ADKAR^®^) change management model, and farm biosecurity was assessed with the Biocheck.UGent^™^ tool. Subsequently, farmers were coached to improve disease prevention and antimicrobial stewardship. After the individual coaching of farmers, there was a change in their attitudes regarding AMU, reflected by an increase in ADKAR^®^ scores. Biosecurity levels improved by around 6% on average, and AMU was reduced by 7% on average without negative effects on performance parameters. Despite these improvements, no significant association could be found between higher ADKAR^®^ scores and lower AMU. Further investigation into sociological models is needed as a tool to reduce AMU in livestock production.

## 1. Introduction

Prudent use of antimicrobials (AMs) is of utmost importance, as it has been shown that increased selection pressure due to the use of AMs is the main driver in the selection of resistant microorganisms [1,2,3,4]. Antimicrobial resistance (AMR) is a global health problem in both human and veterinary medicine. In particular, in broiler production (which represents approximately 37% of global animal protein production [5]), antimicrobial use (AMU) is relatively high, and elevated levels of AMR have been found in comparison to other food-producing animals [6,7,8].

To control the problem of AMR and the possible exchange of resistant microorganisms between animals and humans [9,10], efforts are being made on both a national and global level to reduce AMU. Monitoring of AMU in combination with regulations has been shown to be effective up to a certain level [11,12,13,14,15]; however, further efforts to reduce AMU are necessary.

Previous studies have shown that the level of AMU on a farm is influenced both by the health status of the animals and the attitudes of the farmers [16,17]. Hence, both need to be addressed if AMU is to be reduced. Farm biosecurity is the combination of all measures taken to prevent the introduction and spread of infectious pathogens on a farm [18]. By improving biosecurity, the occurrence of disease can be prevented, and the need for AMU can be lowered [2,19,20,21,22]. Farmers’ willingness to embrace changes also plays an important role in improving biosecurity, as changes in daily routine are required and the benefits are not always noticeable from the start [23]. Therefore, improving biosecurity and reducing AMU requires changes in the attitudes and behaviors of farmers.

As achieving a sustained change in behavior is difficult [24], coaching may be helpful. Coaching is the process of helping individuals to identify their goals through non-directive questioning and interaction. It differs from the conventional advising process (traditionally undertaken by veterinary experts) which is mainly focused on responding to a specific question or event and is more unidirectional [25].

Individual guidance and the coaching of farmers have already proven their effects in pig production [22,26,27]. However, in these studies coaching was performed intuitively, and it was difficult to link the obtained results to the specific personality traits of the farmers and/or the effects of a change in attitude. Therefore, in the current study we used the Awareness, Desire, Knowledge, Ability, and Reinforcement (ADKAR^®^) change management model, after adaptation for use in veterinary medicine [28], to assess farmers’ attitudes and behavior regarding AMU and infection prevention. The ADKAR^®^ model is an acronym for the building blocks required for successful behavior change. Farmers were scored for each building block from 1 to 5, with 1 indicating the greatest barriers to change.

This study aimed to assess to what extent improvements in farm biosecurity, reductions in AMU, and improved attitudes towards antimicrobial stewardship could be obtained in 30 broiler farms in Belgium and the Netherlands through coaching and, subsequently, how these improvements could be linked to the ADKAR^®^ profiles of the farmers.

## 2. Results

The main characteristics of the participating farms are described by Caekebeke et al. [29]. Except for a higher average number of broilers per farm in the Netherlands, participating farms in both countries showed similar characteristics.

### 2.1. Evolution of Farm Parameters

#### 2.1.1. ADKAR^®^ Scores

Based on a semi-structured interview, farmer-specific ADKAR^®^ profiles were established to quantify attitude changes in three different periods: at the start of the study (period 1); after 6 months of coaching (period 2); and at the end of the study (after 12 months of coaching). The results from the participating farmers are presented in Table 1. Two farms in the Netherlands withdrew from the study.

At the country level, each of the elements either increased (mainly when they began at relatively low levels) or remained the same (mainly when they began at already high levels) from periods 1 to 3.

#### 2.1.2. Farm Interventions

Based on the specific situation in each farm, improvements in biosecurity and overall farm management were intended to reduce the need for AMU. These improvements were divided into different categories, as can be seen in Table 2. The majority of the improvements are related to farm hygiene, the quality of the drinking water, and the management of 1-day-old chicks. A non-exhaustive list of suggested improvements is also included.

The improvements shown in Table 2 were suggested at farm visits 2 and/or 3, as some farms were already able to make changes from the start, while others needed more time.

#### 2.1.3. Biosecurity

Average levels of both external and internal biosecurity increased from period 1 to 3 in Belgian and Dutch farms (Figure 1). External biosecurity levels were (± STD), on average, higher (64 ± 7, and 71 ± 6 in Belgian and Dutch farms, respectively) than internal biosecurity (58 ± 11, and 68 ± 7 in Belgian and Dutch farms, respectively) during the study period.

The Biocheck.UGent™ subcategory infrastructure and biological vectors (external biosecurity) scored best in both countries, meaning that most farms had a good control program against vermin and wild animals, e.g., fencing of the farm, grids placed on air inlets, and an effective rodent control program. The subcategory in need of the most attention was feed and water (average of 45 (±6) and 58 (±15) for Belgian and Dutch farms, respectively). These low scores were mainly due to the limited number of farms that regularly tested water quality and the high frequency of feed delivery trucks coming to the farm.

#### 2.1.4. Antimicrobial Use

All AMs were administered as group treatments via the drinking water. The median treatment incidence (TI) for the Belgian farms at the start of the project (period 1) was 8.16 (range = 0.00–47.11; $\bar{\begin{array}{c} x \end{array}}\;$= 10.32). This corresponds to AM treatment durations of 3.4 days for a production round of 42 days on average. Antimicrobial use decreased to 8.15 (range = 0.00–52.91; $\bar{\begin{array}{c} x \end{array}}$ = 10.80) and 7.25 (range = 0.00–49.12; $\bar{\begin{array}{c} x \end{array}}$ = 10.18) in coaching periods 2 and 3, respectively. This corresponds to a total reduction of 11%. The participating broiler farms in the Netherlands had a median TI of 2.69 (range = 0.00–44.65; $\bar{\begin{array}{c} x \end{array}}$ = 6.06) at the start, of 4.51 (range = 0.00–29.80; $\bar{\begin{array}{c} x \end{array}}$ = 5.67) in the second observation period, and of 2.64 (range = 0.00–27.24; $\bar{\begin{array}{c} x \end{array}}$ = 5.92) at the end. This results in an overall reduction of 2% from periods 1 to 3. Figure 2 shows the AMU for both countries during the study period.

During the whole study, the majority of AMs used were classified as second-choice substances (76%). Extended-spectrum penicillin (amoxicillin) and a combination product containing lincomycin with spectinomycin (Belgium) were the main AMs used. Although the amount of critically important AMs was already low at the start of the study (1% of the total amount of AMs used on average), a further decline throughout the study was observed (0.2% in period 3).

#### 2.1.5. Technical Performance Parameters

On average, mortality declined in Belgian farms during the study. Farms had a median mortality in period 1 of 2.95% (range = 1.44–7.07%), which declined to 2.90% in period 2 (range = 1.29–6.08%) and 2.28% in period 3 (range = 1.29–5.30%). The Dutch farms showed an increase in mortality between period 1 (median = 2.95%; range = 1.16–5.40%) and 2 (median = 3.29%; range = 1.99–7.42%), with a slight decrease in period 3 (median = 3.16%; range = 1.85–7.81%).

In both countries, the feed conversion ratio (FCR) declined over time from 1.59 and 1.58 in Belgian and Dutch farms, respectively, to 1.56 in both countries.

### 2.2. Associations between Parameters

Significant associations according to the linear mixed model are depicted in Figure 3. The borderline significant associations between Ability and country with AMU were included as well. All results are corrected with the individual farm as a random factor.

Estimates are relative to the baseline level of coaching period 1 in Belgium for low ADKAR^®^ scores.

No significant associations between the individual ADKAR^®^ elements and AMU could be found. For Table 3, all participating farms per country were grouped by their score (low or high) for the different ADKAR^®^ elements. Subsequently, from those farms, we looked at whether they had an AMU below, equal to, or above the country-specific median value within the different coaching periods.

## 3. Discussion

In this study, we quantified attitudes towards disease prevention and antimicrobial stewardship in 28 broiler farmers in Belgium and the Netherlands through an adjusted ADKAR^®^ change management model. Farmers were coached to improve infection prevention and reduce AMU based on the results of their ADKAR^®^ profiling. Despite the fact that the on-farm interactions between biosecurity, antimicrobial use, production parameters, and farmers’ ADKAR^®^ profiles were found to be complex (Figure 3), the results indicate some relevant associations. No association between the ADKAR^®^ elements and AMU could be found significant in our study.

To the best of our knowledge, this is the first time that attitudes towards antimicrobial stewardship in veterinary medicine have been quantified using the ADKAR^®^ model, with the exception of the methodology described in [28]. We also included two cycles of intervention, whereas other studies only had one follow-up period [26,30].

Despite the fact that all data collected in period 1 were not influenced by interventions (meaning that period 1 can be seen as a control period), the presence of time effects cannot be ruled out as we did not include concurrent control farms in our study. In addition, due to the selection criteria, the farms included are not representative of broiler production in Belgium and the Netherlands.

Initial profiling according to the adapted ADKAR^®^ model in period 1 showed relatively low scores for all elements in Belgian farmers, suggesting the presence of an existing barrier to achieving change, as described by Hiatt [31]. This indicated the need for general improvements in all elements. Participating Dutch farmers showed higher scores for most elements at the beginning of the study, with the exception of Knowledge.

As the initial ADKAR^®^ scores for Dutch farmers were already high for some elements (with a maximum score of 5), there was little room for further improvement throughout the study. Belgian farmers, who had more room for improvement, showed an increase in scores for all ADKAR^®^ elements throughout the study.

The higher overall scores in Dutch farmers could be the result of the earlier initiation of benchmarking of AMU and definition of AMU reduction goals in the Netherlands (6 years earlier than in Belgium) [11,32], creating awareness and the need for antimicrobial stewardship. Importantly, although Dutch farmers seem more aware of the problem (Awareness), and want to act upon this (Desire), more knowledge on how certain aspects could be changed was needed at the start of the project. Moreover, differences in ADKAR^®^ profiles between both countries may indicate different perceptions of antimicrobial stewardship, suggesting that behavioral change requires a different approach in both countries.

Over time, biosecurity levels significantly improved. However, more room for improvement was found in Belgian farms (lower average scores). This might explain the larger improvements seen in Belgian farms, as it is more difficult to make changes on farms that already have well-established biosecurity protocols.

In both countries, internal biosecurity levels scored lower than levels of external biosecurity. This may be due to a greater awareness of the risk of introduction of disease as a consequence of large outbreaks of infectious diseases in the past (such as highly pathogenic avian influenza [33,34]), or the belief that it is easier to impose guidelines upon external visitors than to change habits on the farm [35]. However, Van Limbergen et al. [36] found internal biosecurity levels to exceed external biosecurity levels in broiler production. This difference could be explained by the inclusion of five European countries in the study of Van Limbergen et al. (including Belgium, but not the Netherlands), together with the specific selection criteria that we handled in our study. Nonetheless, individual farms in both countries with high external biosecurity levels throughout the study also had a higher internal biosecurity level on average. This association was described in a previous study [34,35,36].

Antimicrobial use decreased throughout the study period in both countries (7% on average), although not significantly. Dutch farms had a lower AMU on average in comparison to Belgian farms, which could, again, be explained by the general policy measures regarding AMU in the Netherlands, creating greater awareness of antimicrobial stewardship and resulting in a lower average use of AMs.

The use of critically important AMs was already low at the start of the project. Coaching helped to further reduce this amount (from 1% to 0.2%). The proportion of first- and second-choice products remained similar throughout the study period, indicating that there was no clear shift from second- to first-choice products.

If a reduction in AMU would have resulted in an inferior performance among the animals, coaching would have been more challenging and long-term changes would have been more difficult to achieve. However, coaching of the participating farmers towards improved biosecurity and reduced AMU did not have negative effects on performance parameters, as both mortality (Belgium) and FCR decreased over time. Moreover, a strong positive association between AMU and mortality was found, with a lower mortality rate being associated with a lower AMU. This may indicate that AMU occurs in the case of disease events or to curb mortality.

An association between biosecurity and AMU, as found in pig production [20], was not found. This, again, indicates that AMU is influenced by other factors, such as the attitude of the farmers and potentially also management, vaccination, or animal genetics [20,34].

Farmers with higher Ability scores linked to the presence of resources such as money, workers, or competencies showed a tendency towards a lower AMU. As almost all Dutch farmers showed high scores for Ability in all coaching periods, collinearity was present between Ability and the Netherlands, likely explaining the absence of a significant association. This is, again, indicative of possible associations between farmer attitudes and AMU evolution when farms begin at a higher level of AMU and a lower Ability, which is typically the case at the beginning of a change process. When farmers already have a high Ability level and a low AMU, as was the case in the Netherlands, coaching for a reduction in AMU might be less efficient, but could focus more on antimicrobial stewardship or disease prevention. However, this study did not allow for the quantification of these relationships.

From Table 3, no clear trends are visible. This could be an explanation for not finding significant associations between ADKAR^®^ and AMU. Moreover, from this table, the limited study size is apparent. Although no significant associations were found, coaching can have relevant improvements on the individual farm level.

The results from this study may be of relevance for future (meta-analysis) studies, with a larger sample size if possible, to further investigate the use of sociological models on AMU reductions in field conditions.

## 4. Materials and Methods

### 4.1. Study Design

A longitudinal study was performed on 15 broiler farms in Belgium and 15 broiler farms in the Netherlands between September 2017 and May 2019. During a 13-month period, participating farms were visited 4 times. Data were collected during 3 of these farm visits.

During the first farm visit, information was collected regarding biosecurity levels, farm characteristics, technical performances, and AMU. Data on technical performance and AMU were retrieved from 365 days preceding the first farm visit.

A second farm visit was planned about 1 month later. During this visit, a farm-specific action plan was discussed, taking into account the farmers’ ADKAR^®^ profiles (see Section 4.3 Coaching methodology). The visit consisted of a face-to-face interview with the farmer and preferably also the herd veterinarian, where positive features of the farm were discussed as well as points of attention. All aspects where the farmer was able and willing to make improvements in the following 6 months were written down as part of an action plan. After 6 months, the action plan was reviewed with the farmer and herd veterinarian. Achievements were recorded and future actions were reconfirmed or adjusted for the following 6 months when deemed necessary. This second improvement period was followed by a fourth and final farm visit. The farmer was provided with feedback on the execution of the action plan and further suggestions were discussed. During the third and fourth visits, farm data were collected again for the period between visits 1 and 3 and the period between visits 3 and 4, respectively. These data were always considered when reviewing the action plan.

All farm visits were executed by 2 researchers/veterinarians: 1 in Belgium and 1 in the Netherlands. Both were trained in coaching by a professional institute and performed 10 mutual farm visits to align their methodologies regarding the collection of data and the interpretation of the biosecurity questionnaire. The researchers kept frequent contact to review and discuss the outcomes and progress.

### 4.2. Farm Selection

In both Belgium and the Netherlands, farms were recruited according to the methodology described in Caekebeke et al. [29]. In brief, the selection criteria were (1) farm type: conventional broiler farms; (2) farm location: within the Belgian–Dutch border region; and (3) AMU: high users according to the national benchmark value. Herd veterinarians were involved as much as possible in order to increase compliance with the action plan and encourage long-term involvement.

Before enrolment, all participants were informed of the aim of the study. All farmers signed a form providing their informed consent to participate in coaching and agreeing to data collection, exchange, and analysis, as well as the publication of the results. The Animal Welfare Bodies from Utrecht University and Ghent University were consulted and concluded that the study was exempt from an animal ethical evaluation, as the project did not include procedures according to EC/2010/63.

In the Netherlands, 2 broiler farms withdrew during the project.

### 4.3. Coaching Methodology

To quantify farmers’ attitudes towards a reduction in AMU, the ADKAR^®^ change management model was used. This model is already well established in the fields of corporate business and human medicine [31,37,38] and offers a practical tool for guiding change. ADKAR^®^ is an acronym for the 5 building blocks necessary for successful change: Awareness, Desire, Knowledge, Ability, and Reinforcement.

The model was adapted by the authors for use in veterinary medicine, where the attitude of a farmer towards AMU was scored for each element. Details on the scoring methodology can be found in [28].

Each building block was scored between 1 and 5, with a score of 3 or less indicating a barrier to possible change, as described by Hiatt [31]. During farm visits 1, 3, and 4, farmers’ profiles were scored according to the ADKAR^®^ principle, based on a semi-structured interview. The Reinforcement element was not scored during farm visit 1 as no measures were implemented yet. Although it was scored during visits 3 and 4, the authors decided not to include this element in the final analyses as we could not cover the entire study period.

The farmer-specific profile determined during farm visit 1 was the starting point for the coaching of the farmers. Whenever the awareness of a farmer regarding a reduction in AMU required attention (low scores), the focus point was to address the “why” question in order to achieve change. This was done by providing information and examples on the risks of high AMU, and why change is important. If farmers scored highly for awareness from the beginning, the “why” issue was only touched upon very briefly. The Desire element is intrinsic to the individual and his/her motivation. Desire could be improved by focusing on the benefits and potential gains for the participating farmer. When farmers had the desire to change, they may have needed help with the “how” part (low scores for Knowledge). This element was addressed by providing detailed information and education on a specific or generic problem, depending on the farm. Low Ability scores were tackled by discussing big investments or structural changes and were divided into smaller and easier steps to help farmers accomplish their goals. Farmers were followed-up and supported. Reinforcement, the final element, was strengthened by focusing on the positive changes throughout the study.

To increase statistical power, the ADKAR^®^ components were dichotomized into low (score 1, 2, or 3) or high (score 4 or 5) categories.

### 4.4. Data Collection

The level of biosecurity was quantified using the Biocheck.UGent™ questionnaire, which is a risk-based scoring system [39] that has been used in numerous studies [21,36,40,41,42]. The Dutch version of the questionnaire for broilers was used, containing 79 questions within multiple subcategories, resulting in a farm-specific score for external (all measures taken to prevent the introduction of pathogens to the farm) and internal (all measures taken to prevent the spread of pathogens within the farm) biosecurity. Scores range from zero to 100, with zero corresponding to the absence of any biosecurity measures and a score of 100 corresponding to the ideal biosecurity situation according to the Biocheck.UGent™ questionnaire.

The questionnaire was filled out on paper with the farmer after visual appraisal of the farm. After each farm visit, the questionnaire was re-entered online (www.biocheck.ugent.be, accessed on 20 April 2021) to generate the biosecurity scores.

Antimicrobial use data were obtained from the farmer or herd veterinarian through registration documents. All AM were classified into first-, second-, and third-choice substances, according to their importance for human medicine, based on the recommendations of the WHO [43]. Quantification was standardized, using the TI per 100 days, as described by Caekebeke et al. [29]. The TI represents the number of days an animal was treated with AM per 100 days or the percentage of treatment days with the following formula:(1)$$\mathrm{TI}=\frac{\mathrm{Total}\;\mathrm{amount}\;\mathrm{of}\;\mathrm{active}\;\mathrm{substance}\;\left(\mathrm{mg}\right)}{\mathrm{DDDvet}\;\left(\frac{\frac{\mathrm{mg}}{\mathrm{kg}}}{d}\right)\times \mathrm{observation}\;\mathrm{period}\;\left(d\right)\times \mathrm{animals}\;\mathrm{at}\;\mathrm{risk}\;\left(\mathrm{kg}\right)}\times 100$$

The numerator represents the total amount of active substances prescribed within a farm. The denominator is the multiplication of (1) the DDDvet (defined daily dose): defined doses of an AM; (2) the observation period: the length of a production period in days; and (3) the number of animals at risk of AM treatment in kg. The latter is the result of multiplying the standard weight for a broiler (1 kg), according to the European Surveillance of Veterinary Antimicrobial Consumption (ESVAC) [44], with the number of broilers on the farm. The entire ratio is then multiplied by 100 animal-days at risk to obtain the TI. By using this standardized formula, AMU within both countries could be compared.

Farm characteristics (number of houses with the total amount of broilers present) and technical performances (mortality and feed conversion ratio) were collected from the farmer and through the farm management programs.

The data in the results section are reported in periods. For technical performance and AMU data, period 1 includes data from 365 days preceding the first farm visit until farm visit 1. Period 2 includes all data between visit 1 and visit 3, period 3 covers the data between visits 3 and 4. Biosecurity levels and ADKAR^®^ scores were determined during the farm visit (visits 1, 3, and 4).

### 4.5. Data Analysis

The data are represented as the mean, median, and the minimum to maximum range for all farms per coaching period.

All dependent variables were analyzed using a linear mixed effects model. The independent variables that were considered as fixed factors are given in Table 4. The farm was set as a random factor. When the outcome variables were available at the production round level, the period was nested in the farm level to take into account the repeated nature of the measurements. Next, we investigated if a simple homoscedastic residual variance was justified, or whether a more complex residual variance was needed. Single fixed factors were added to the model and assessed for significance. Following this, all significant factors were withheld for use in the model. Subsequently, a multivariate model was built in a forward, stepwise manner. Each time a factor was added, significance was tested with a type-III ANOVA test. In addition to the main effects, 2-way interactions with country were also assessed.

All statistics were performed using R version 4.0.2 (R core team, Vienna, Austria), with models fitted using the R nlme package [45].

## 5. Conclusions

After individual coaching of farmers there was a change in attitude and behavior regarding AMU, as reflected by an increase in ADKAR^®^ scores. In addition, biosecurity levels and AMU improved following coaching, without negative effects on performance parameters. The results from this study indicate the positive effects of coaching on farm performance and antimicrobial stewardship. Further investigation into the use of sociological models such as the ADKAR^®^ model is needed as a tool for improved antimicrobial stewardship.

## Figures and Tables

**Figure 1 antibiotics-10-00590-f001:**
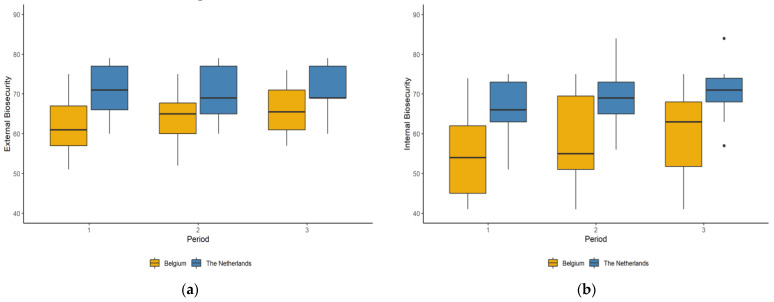
Distribution of (**a**) external and (**b**) internal biosecurity levels for participating Belgian (*n* = 15) and Dutch (*n* = 13) farms per coaching period. The higher the biosecurity scores are on the *y*-axis, the better the disease prevention measures in place are. The line within each box represents the median value. The lower and upper boundaries of each box correspond to the 25th and 75th percentiles, respectively. Whiskers below and above each box show the 10th and 90th percentiles. Points below and above the whiskers represent outliers.

**Figure 2 antibiotics-10-00590-f002:**
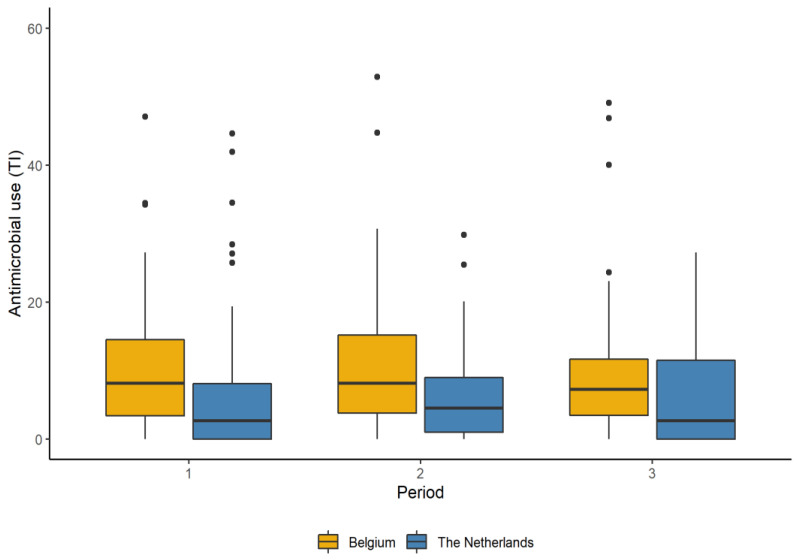
Antimicrobial use per coaching period for the participating broiler farms in Belgium (*n* = 15) and the Netherlands (*n* = 13). Antimicrobial use is expressed as the number of days an animal was treated with antimicrobials out of 100 days (treatment incidence (TI) per 100 days). The line within each box represents the median value. The lower and upper boundaries of each box correspond to the 25th and 75th percentiles, respectively. Whiskers below and above each box are the 10th and 90th percentiles. Points below and above the whiskers represent outliers.

**Figure 3 antibiotics-10-00590-f003:**
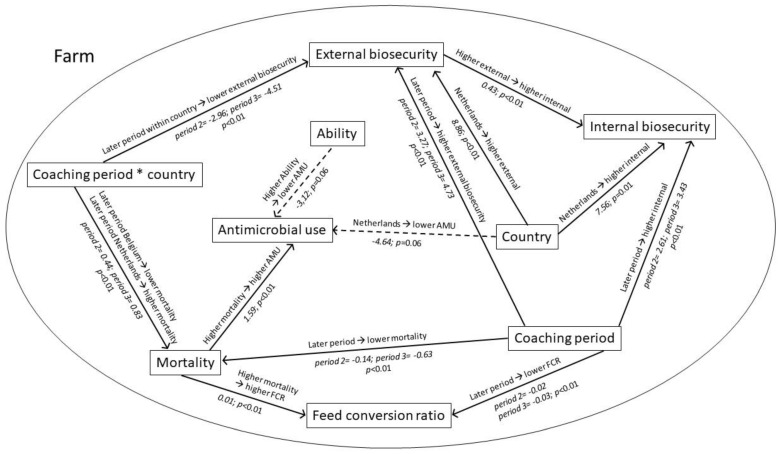
Association pathway according to a linear mixed model. The model included country, coaching period, biosecurity levels, performance parameters (mortality, feed conversion ratio), ADKAR^®^ elements, and antimicrobial use. AMU: antimicrobial use; FCR: feed conversion ratio. Coaching period * country: the interaction between coaching period and country. The model was corrected for farm effects by assigning the latter as a random factor. Estimates were added to the pathway in italics. The non-significant associations with AMU are indicated with a dashed arrow.

**Table 1 antibiotics-10-00590-t001:** Awareness, Desire, Knowledge, Ability, and Reinforcement (ADKAR^®^) scores of the broiler farmers for all coaching periods. Reinforcement (R) was not determined in period 1.

Belgium (*n* = 15)	**Period**		**A**	**D**	**K**	**Ab**	**R**
	1	Mean	2.93	3.33	3.00	2.67	NA
		Median	3.00	3.00	3.00	2.00	
	2	Mean	3.20	3.20	3.67	3.13	2.60
		Median	4.00	4.00	4.00	3.00	4.00
	3	Mean	3.47	3.53	3.13	3.20	3.27
		Median	4.00	4.00	3.00	3.00	4.00
The Netherlands (*n* = 13)	**Period**		**A**	**D**	**K**	**Ab**	**R**
	1	Mean	4.80	4.40	2.40	4.10	NA
		Median	5.00	5.00	2.00	4.00	
	2	Mean	4.78	4.67	2.78	4.44	4.78
		Median	5.00	5.00	3.00	4.00	5.00
	3	Mean	4.86	4.43	3.71	4.29	4.57
		Median	5.00	5.00	4.00	4.00	5.00

A: Awareness; D: Desire; K: Knowledge; Ab: Ability; R: Reinforcement; NA: not applicable. Period 1: Scores at the start of the study period (farm visit 1); period 2: scores after 6 months of coaching (farm visit 3); period 3: scores after 12 months of coaching (farm visit 4).

**Table 2 antibiotics-10-00590-t002:** The percentage of farms in Belgium and the Netherlands where the different improvement categories were suggested.

Category	Improvements	Participating Farms (%)
Hygiene	The layout of the hygiene lock, farm-specific clothing and footwear for the catching team, more regular washing of hands, vehicles should always be empty upon arrival	61
Quality of the drinking water	More frequent testing of the drinking water and analysis of the results, thorough cleaning and disinfection of the drinking lines within the houses	54
Management 1-day-old chicks	Always purchasing from the same hatchery, reducing the amount of time between hatching and transportation to the farm, appropriate floor temperature in the houses	43
Infrastructure of the farm	Having dedicated clean and dirty areas on the farm, grids in front of air inlets	36
Treatments	Improvements to vaccination schemes, more prudent use of antimicrobials	36
Follow-up of farm data	Evaluation of past rounds, receiving post-mortem reports from the slaughterhouse	18
Quality of the feed	Improving feed composition, adding less homegrown crops	7
Management of the farm	No partial depopulation, all-in/all-out	4

**Table 3 antibiotics-10-00590-t003:** The number of farms per country and coaching period are grouped according to their score (low/high) for each ADKAR^®^ element and their antimicrobial use below or equal to/above the country-specific median value.

	**Period**	**TI Relative to Median**	**A**		**D**		**K**		**Ab**	
			**Low**	**High**	**Low**	**High**	**Low**	**High**	**Low**	**High**
Belgium (*n* = 15)	1	<	4	4	4	4	5	3	6	2
		≥	6	1	4	3	6	1	5	2
	2	<	3	5	2	6	2	6	4	4
		≥	4	3	3	4	4	3	5	2
	3	<	4	4	3	5	4	4	5	3
		≥	3	4	4	3	6	1	4	3
	**Period**		**A**		**D**		**K**		**Ab**	
			**Low**	**High**	**Low**	**High**	**Low**	**High**	**Low**	**High**
The Netherlands (*n* = 13)	1	<	0	7	1	6	5	2	0	7
		≥	0	6	1	5	5	1	1	5
	2	<	1	6	1	6	5	2	0	7
		≥	1	5	2	4	4	2	0	6
	3	<	0	6	0	6	0	6	0	6
		≥	1	5	1	5	2	4	1	5

In period 3, from 1 Dutch farm no ADKAR^®^ score could be retrieved. A: Awareness; D: Desire; K: Knowledge; Ab: Ability. Period 1: Start of the study period (farm visit 1); period 2: After 6 months of coaching (farm visit 3); period 3: After 12 months of coaching (farm visit 4). TI: treatment incidence per 100 days or the number of days an animal was treated with antimicrobials out of 100 days.

**Table 4 antibiotics-10-00590-t004:** Overview of the relationships evaluated in this study with different dependent, fixed, and random variables.

Dependent Variable	Independent Variables (Fixed)	Independent (Random)
AMU	Awareness, Desire, Knowledge, Ability, external biosecurity, internal biosecurity, mortality, period, country	Farm, period in farm
Mortality	Awareness, Desire, Knowledge, Ability, external biosecurity, internal biosecurity, AMU, period, country	Farm, period in farm
FCR	Awareness, Desire, Knowledge, Ability, external biosecurity, internal biosecurity, AMU, period, country, mortality	Farm, period in farm
External biosecurity	Awareness, Desire, Knowledge, Ability, internal biosecurity, period, country	Farm
Internal biosecurity	Awareness, Desire, Knowledge, Ability, external biosecurity, period, country	Farm

FCR: feed conversion ratio; AMU: antimicrobial use.

## Data Availability

The data presented in this study are available on request from the corresponding author. The data are not publicly available due to agreements made within the i-4-1-Health consortium to make the i-4-1-Health datasets publicly available no later than 31 December 2024, following the FAIR (Findable, Accessible, Interoperable, Reusable) data principles.

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
