# Peer review of "Coaching Belgian and Dutch Broiler Farmers Aimed at Antimicrobial Stewardship and Disease Prevention"

_antibiotics, 2021, doi:10.3390/antibiotics10050590_

Round 1

Reviewer 1 Report

The manuscript has a narrow scope as it is about coaching farmers about prevention of disease. The topic is important but is more behavioral than biological. In addition, the conclusions of the manuscript are not convincing. The authors state that although not significant, the results indicate a relationship between higher ADKAR®-scores 28 and a lower AMU. Without significance, I do not think we can draw any conclusions or make any recommendations. 

Author Response

We would like to show our greatest appreciation to the reviewer for investing the time and energy into reading this manuscript.

Please find our answers in the attachment.

Reviewer 2 Report

Prudent use of antimicrobials or reduction of antimicrobial use (AMU) is very important for livestock especially for broilers. One new model that of ADKAR® change management model and farm biosecurity was used to assess the Biocheck in the manuscript, and it suggests that the coaching of farmers through the ADKAR® model can be a good tool to aid in sustainable AMU reduction. The paper is well designed and written, and the conclusion was reasonable.

Author Response

We would like to thank the reviewer for the feedback. We appreciate the time and effort the reviewer has put into reviewing this manuscript.

Reviewer 3 Report

The study investigated coaching belgian and dutch broiler farmers aimed at antimicrobial stewardship and disease prevention. Social scientific approach for the AMU reduction is interesting, but more information or explanation are needs for the readability. ADKAR is a business developing tool, which may be familiar for most of the researcher. Therefore, the brief explanation for the tool is required before Results. Is there the similar investigation using this tool in agriculture study? If so, it should be introduced.

Abstract

  1. Line 30; Please consider to revise “sustainable” into

Introduction

  1. Line 48-51; Rewrite the sentence without using ( ) because the content was too long.

Results

  1. Table 2; Show more information of each parameter in Improvements, e.g., deeper breeding sterilization or frequent changes in antiseptic solution for Hygiene. Resuls also should be present before Result.

Author Response

The authors would like to thank the reviewer for the valuable feedback.

Please find our answers in the attachment. 
